# High hydrostatic pressure stimulates microbial nitrate reduction in hadal trench sediments under oxic conditions

Na Yang [1], Yongxin Lv [1], Mukan Ji[2], Shiguo Wu[3] & Yu Zhang [1,4,5] ✉

Hadal trenches are extreme environments situated over 6000 m below sea surface, where enormous hydrostatic pressure affects the biochemical cycling of elements. Recent studies have indicated that hadal trenches may represent a previously overlooked source of fixed nitrogen loss; however, the mechanisms and role of hydrostatic pressure in this process are still being debated. To this end, we investigate the effects of hydrostatic pressure (0.1 to 115 MPa) on the chemical profile, microbial community structure and functions of surface sediments from the Mariana Trench using a Deep Ocean Experimental Simulator supplied with nitrate and oxygen. We observe enhanced denitrification activity at high hydrostatic pressure under oxic conditions, while the anaerobic ammonium oxidation – a previously recognized dominant nitrogen loss pathway – is not detected. Additionally, we further confirm the simultaneous occurrence of nitrate reduction and aerobic respiration using a metatranscriptomic dataset from in situ RNA-fixed sediments in the Mariana Trench. Taken together, our findings demonstrate that hydrostatic pressure can influence microbial contributions to nitrogen cycling and that the hadal trenches are a potential nitrogen loss hotspot. Knowledge of the influence of hydrostatic pressure on anaerobic processes in oxygenated surface sediments can greatly broaden our understanding of element cycling in hadal trenches.

The hadal trenches are deep-sea ecosystems (depth range of 6000–11,000 m) featured by high hydrostatic pressure ($\geq 60$ MPa)[1,2]. Despite their extreme conditions, hadal trenches are hotspots of elemental cycling with high organic matter deposition rate and microbial activities[3–6]. Recent investigations, grounded in both experimental evidence and modeling results, have revealed that bioavailable nitrogen species (such as $NH_4^+$, $NO_2^-$, and $NO_3^-$) are converted into biologically inert $N_2$ in hadal sediments[7–10]. This continuous loss of nitrogen leads to an elevated carbon-to-nitrogen ratio and even nitrogen-limitation to benthic microorganisms[11,12], which subsequently alters the biogeochemical processes in the deep ocean floor. Thus, a

comprehensive understanding of the mechanism of nitrogen loss is vital in deciphering the biogeochemistry of this unique environment.

Denitrification and anaerobic ammonium oxidation (anammox) are the major known microbial processes that produce $N_2$. These processes are generally considered anaerobic[13,14], and genes involved in these processes have been identified in various trench sediments, irrespective of oxygen availability[7,15,16]. Currently, anammox is recognized as the dominant process responsible for nitrogen loss in anoxic sediments, contributing significantly more than denitrification. For instance, Thamdrup et al. studied the bottom-axis sediments from Atacama Trench and Kermadec Trench, demonstrating the

[1]School of Oceanography; Shanghai Key Laboratory of Polar Life and Environment Sciences; MOE Key Laboratory of Polar Ecosystem and Climate Change, Shanghai Jiao Tong University, Shanghai, China. [2]Center for Pan-third Pole Environment, Lanzhou University, Lanzhou, China. [3]Institute of Deep-sea Science and Engineering, Chinese Academy of Science, Sanya, China. [4]Laboratory for Polar Science, Polar Research Institute of China, Ministry of Natural Resources, Shanghai, China. [5]Yazhou Bay Institute of Deepsea Sci-Tech, Shanghai Jiao Tong University, Sanya, China. ✉e-mail: zhang.yusjtu@sjtu.edu.cn

prominence of anammox, while denitrification generally occurred at lower rates and was limited to the surface layer[8]. Zhou et al. reported the heterogenous distribution of denitrification and anammox across the Mariana Trench, with the anammox being more intensive in bottom-axis sediments with stronger oxygen depletion compared to the slope sites[9]. However, the contribution of denitrification on nitrogen loss in oxygenated trench surface sediments is yet to be investigated.

Microbial physiology, activities, and metabolism in hadal trenches are affected by hydrostatic pressure[1,17]. Under high hydrostatic pressures, microorganisms prefer anaerobic metabolisms over aerobic respiration. This is explained by the "common adaptation strategy", which speculates that anaerobic metabolisms cause less intracellular oxidative stress than the latter[17–19]. Furthermore, high hydrostatic pressures shift the redox ladder, leading to a greater reaction of Gibbs free energy being generated from denitrification compared to that under ambient pressure[20]. Therefore, we propose a hypothesis that in hadal trench sediments where the hydrostatic pressure is extremely high, denitrification is an energetically favorable pathway contributing to the nitrogen loss in the oxygenated upper layer sediment. In this study, we test this hypothesis using a specially-designed Deep Ocean

Experimental Simulator to ensure a continuous supply of oxygen, using the Mariana Trench sediments as the inoculum (Fig. 1). We then apply microbial activity analysis, metagenomic and metatranscriptomic analyses to decipher the mechanisms behind it.

## Results
### High hydrostatic pressures modified microbial community trajectory
To investigate the influence of hydrostatic pressure on the microbial community and functions, the sediment sample, collected at a water depth of 6002 m in the Mariana Trench, was sequentially incubated at 0.1, 40, 70, 90, and 115 MPa for 15 days each with a continuous supply of nitrate and dissolved oxygen. With the elevated hydrostatic pressures, the cell numbers of both bacteria and archaea declined (except for a slight increase for bacteria at 70 MPa). Nevertheless, the bacterial population consistently outnumbered the archaeal population by two orders of magnitude (Fig. 2A, B and Supplementary Data 1).

The hydrostatic pressure had distinctive effects on bacterial and archaeal community compositions. For the bacterial community, its taxonomic composition dramatically shifted with increasing hydrostatic pressures: under 0.1 MPa and 40 MPa, *Halomonadaceae* (31.8%

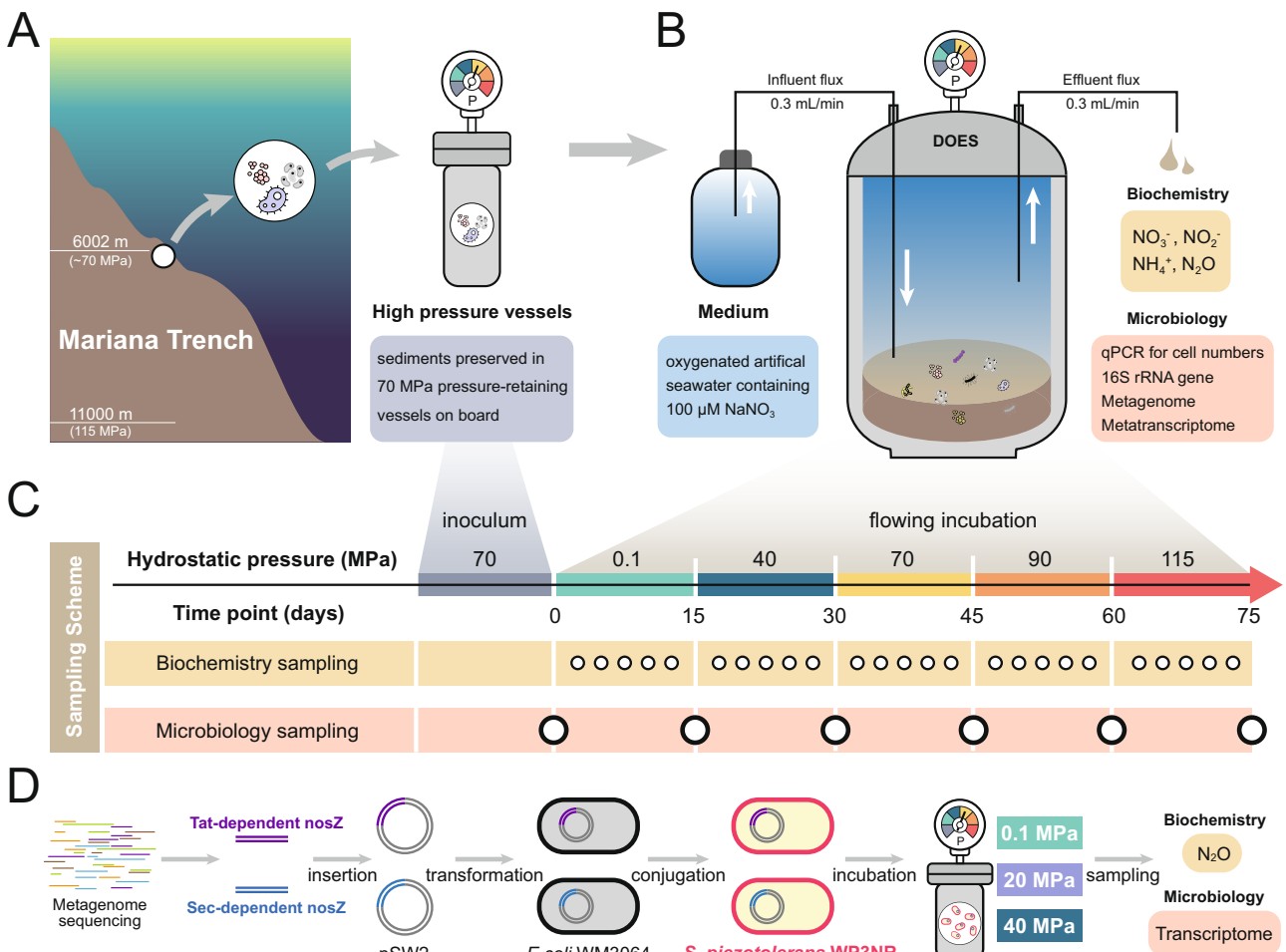

**Fig. 1 | Cartoon shows the overall experimental and analytical design for ascertaining microbial nitrate-reducing pathway with elevated hydrostatic pressures. A** The initial sediment samples were collected in the Mariana Trench, then immediately on board kept in 70 MPa pressure-retaining vessels until used for subsequent experiments. **B** Schematic diagram of continuous flowing incubation for sediments under gradually increasing hydrostatic pressures using Deep Ocean Experimental Simulator (DOES) system that is constantly supplied fresh oxygenic medium with 100 μmol/L of nitrate. **C** The sampling time points for biochemistry and microbiology analyses during continuous flowing incubation under different hydrostatic pressures. **D** For final-step of denitrification, experimental diagram for heterologous expression of the two major clades of *nosZ* genes from sequenced metagenome in this study, Tat-dependent and Sec-dependent in *Shewanella piezotolerans* WP3NR host, and subsequent determination of $N_2O$ reduction activity for the two clades of *nosZ* genes and transcription analysis at 0.1, 20, and 40 MPa.

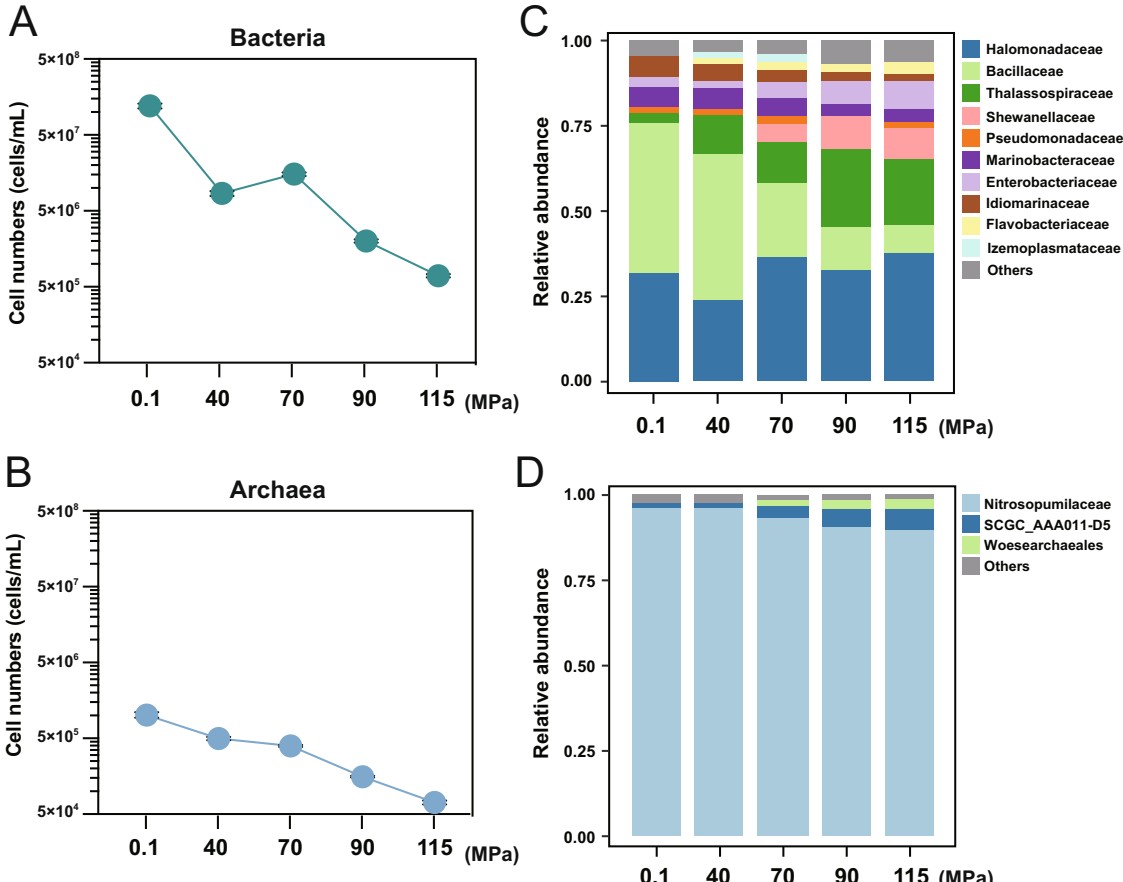

**Fig. 2 | Changes in sediment microbial communities during the continuous hydrostatic pressure flowing incubation.** The calculated cell numbers of bacteria (**A**) and archaea (**B**) are shown as mean values with standard deviation (mean ± sd; $n = 3$) in flowing incubation sediment samples. The family-level summary of bacterial (**C**) and archaeal (**D**) taxonomic assignments for flowing incubation sediment samples under different hydrostatic pressures. Relative abundance of less than 1% is classified as "Others". Source data are provided as a Source Data file.

and 23.6%, respectively) and *Bacillaceae* (44.0% and 43.0%, respectively) were the dominant groups; Under hydrostatic pressures of 70, 90 and 115 MPa, *Halomonadaceae* remained dominant (32.5–37.4%); with the increased relative abundance of *Thalassospiraceae* (11.9%, 23.1% and 19.3%), *Enterobacteriaceae* (4.5, 6.6 and 8.2%), and *Shewanellaceae* (5.4, 9.4 and 9.2%) (Fig. 2C; Supplementary Data 2). In contrast, the archaeal taxonomic composition remained stable under all hydrostatic pressures with the dominance of *Nitrosopumilaceae* (89.5–96.3%) (Fig. 2D and Supplementary Data 2).

### High hydrostatic pressures promote anaerobic nitrate consumption under oxic conditions

We monitored the fluxes of dissolved nitrogen species ($NO_3^-$, $NO_2^-$, and $NH_4^+$) and gaseous nitrogen species ($N_2O$) across the entire incubation. A relatively stable nitrate consumption (38.4–49.5 µmol/day) was observed under a continuous supply of dissolved oxygen (Fig. 3A and Supplementary Data 3). Moreover, because the cell number decreased under elevated hydrostatic pressures, the nitrate consumption rate per cell actually increased (Figs. 2A and 3A). Net nitrite production was also observed, with the rate being significantly higher at 70 MPa (3.00 µmol/d) than those at 0.1 and 40 MPa (one-way ANOVA with Tukey's multiple comparisons test, all $p < 0.05$) (Fig. 3B and Supplementary Data 3).

The mechanism of the enhanced nitrate consumption was investigated using metagenomic and metatranscriptomic analyses. We estimated the gene transcription levels based on the percentage of gene transcripts (relative to all nitrogen transformation-related genes) associated with each pathway of nitrogen cycling (Fig. 4A and Supplementary Data 4). The *napAB* and *narGHI* genes involved in the first step of dissimilatory nitrate reduction were transcribed more actively at higher hydrostatic pressures. Specifically, the transcriptional activity of nitrate reduction to nitrite pathway accounted for 5.2% of all nitrogen transformation-related genes at 0.1 MPa, then increased to 9.6% at 40 MPa, 21.7% at 70 MPa, 12.8% at 90 MPa, and 13.0% at 115 MPa. In additionally, genes involved in the terminal oxygen reduction were abundant and actively transcribed over the entire incubation period (Supplementary Fig. 1).

### High hydrostatic pressure stimulates denitrification but suppresses the DNRA pathway

The $NO_2^-$ produced from $NO_3^-$ reduction has two alternative transformation pathways. It can either be further reduced to NO, $N_2O$, and finally to $N_2$ through denitrification, or to $NH_4^+$ through the dissimilatory nitrate reduction to ammonium (DNRA). The transcription level of all genes involved in denitrification (*nirS*, *nirK*, *norBC*, and *nosZ*) was up-regulated under high hydrostatic pressures (Fig. 4A and Supplementary Fig. 2 and Supplementary Data 4). Specifically, the percentage of *nirS* and *nirK* gene transcripts ($NO_2^- \rightarrow NO$) increased from 2.3% at 0.1 MPa to 15.3% and 12.9% at 90 MPa and 115 MPa, respectively (Fig. 4A). For *norBC* genes ($NO \rightarrow N_2O$), the transcripts increased from 4.0% at 0.1 MPa to 9.9% at 40 MPa, 20.2% at 70 MPa, 28.6% at 90 MPa, but then dropped to 8.7% at 115 MPa. For the *nosZ* gene ($N_2O \rightarrow N_2$), the transcripts increased from 3.1% at 0.1 MPa to 19.3% at 90 MPa and 36.8% at 115 MPa, respectively (Fig. 4A). The enhanced denitrification

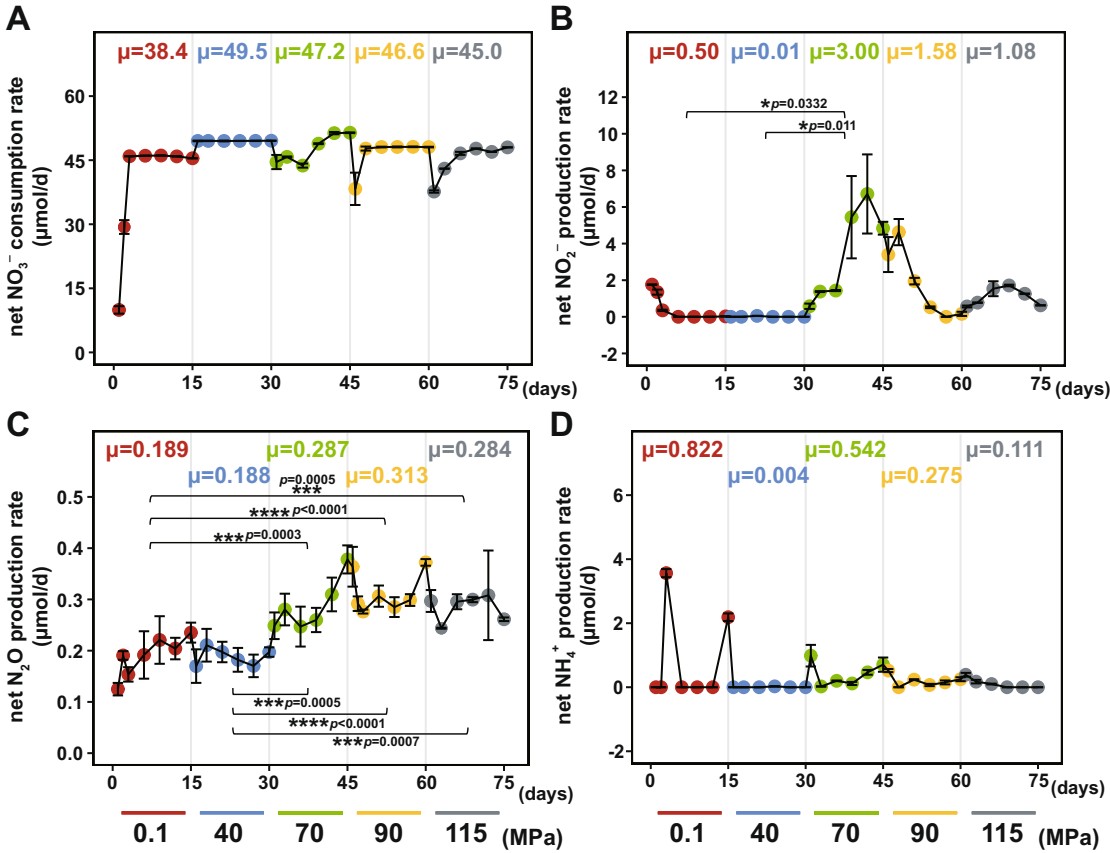

**Fig. 3 | Dot-line plots showing the daily microbial net nitrogen metabolism activity under different hydrostatic pressures.** The calculated net nitrate consumption rate (**A**), net nitrite production rate (**B**), net ammonia production rate (**C**), and net nitrous oxide production rate (**D**) are presented as mean values (μ) with standard deviation (μ ± s.d) based on $n = 7$ (0.1 MPa), $n = 6$ (40 MPa), $n = 7$ (70 MPa), $n = 7$ (90 MPa), $n = 6$ (115 MPa). The $p$ values are analyzed using an ordinary one-way ANOVA with Tukey's multiple comparisons tests. The significant variables are indicated by an overlying line and asterisk (*$P < 0.05$; **$P ≤ 0.01$; *** $P ≤ 0.001$; **** $P ≤ 0.0001$). Source data are provided as a Source Data file.

activity coincided with the strengthened net N₂O production rates at high hydrostatic pressures. Specifically, the N₂O production rate was significantly higher at 70 MPa (0.287 μmol/d), 90 MPa (0.313 μmol/d), and 115 MPa (0.284 μmol/d) than that at 0.1 MPa (0.189 μmol/d) (one-way ANOVA with Tukey's multiple comparisons test, $P < 0.05$) (Fig. 3C and Supplementary Data 3).

As for the DNRA pathway, the chemical profile analysis showed that the average net ammonium production rate at 0.1 MPa was higher than those under high hydrostatic pressures (such as 90 MPa and 115 MPa), although the difference was not significant (one-way ANOVA with Tukey's multiple comparisons test, $p = 0.2629$ (Fig. 3D). The metatranscriptomic analysis showed that the transcription level for the key genes (especially *nirBD*) in DNRA pathway was the highest (80.2%) under 0.1 MPa, which was then halved to 40.9% at 40 MPa and 37.1% at 70 MPa, and then further decreased to 20.7% and 24.5% when the hydrostatic pressure rose to 90 and 115 MPa, respectively (Fig. 4A). Hence, this implies that the activity of DNRA pathway declined with the elevated hydrostatic pressures when the other environmental parameters were kept unchanged.

### Sec-dependent N₂O reductase is selected over Tat-dependent N₂O reductase under high hydrostatic pressures

The transcriptional activity of the *nosZ* gene, which catalyzes the last step of denitrification leading to nitrogen loss through N₂, was elevated under 115 MPa compared with any other hydrostatic pressure used in this study (Supplementary Data 4; Supplementary Fig. 2). Based on the phylogenetic analysis, we identified two types of *nosZ* gene (i.e., Sec-dependent and Tat-dependent, Fig. 5A) in this flowing

incubation system. Most Sec-dependent *nosZ* gene sequences were assigned to Flavobacteriale, whereas the Tat-dependent *nosZ* genes mainly belonged to the Alphaproteobacteria and Gammaproteobacteria (Supplementary Data 5). The metatranscriptomic analysis found a higher fold change with elevated hydrostatic pressures in the transcripts of the Sec-dependent *nosZ* than those of the Tat-dependent *nosZ* (especially at 115 MPa, Sec vs. Tat = 245 vs. 45, Supplementary Fig. 3).

To confirm the preference of Sec-dependent over Tat-dependent *nosZ* genes under high hydrostatic pressures, both genes were synthesized and cloned into the piezotolerant *Shewanella piezotolerans* WP3NR (Fig. 1D). We then performed heterologous gene expression and activity assays of nitrous oxide reductase by quantifying the consumption of the supplied N₂O under various hydrostatic pressures. Our results showed that *S. piezotolerans* WP3NR with the Sec-dependent *nosZ* gene exhibited a significantly higher N₂O consumption rate than that with the Tat-dependent *nosZ* gene (multiple *t*-tests, $p < 0.0001$) (Fig. 5B; Supplementary Data 6a). Additionally, after incubating at 0.1, 20, and 40 MPa for 24 h, the amount of N₂O consumed by the *S. piezotolerans* WP3NR with Sec-dependent *nosZ* gene gradually increased with the elevated hydrostatic pressures, which was the highest at 40 MPa (56.08 μmol). Transcriptomic analysis consistently showed that the Sec-dependent *nosZ* gene was more actively transcribed with increasing hydrostatic pressure, being significantly higher than the Tat-dependent *nosZ* gene (multiple *t* tests, $p < 0.0001$) (Fig. 5C and Supplementary Data 6b). These results confirm that microorganisms prefer to use Sec-dependent N₂O reductase to reduce N₂O and to produce N₂ under high hydrostatic pressures.

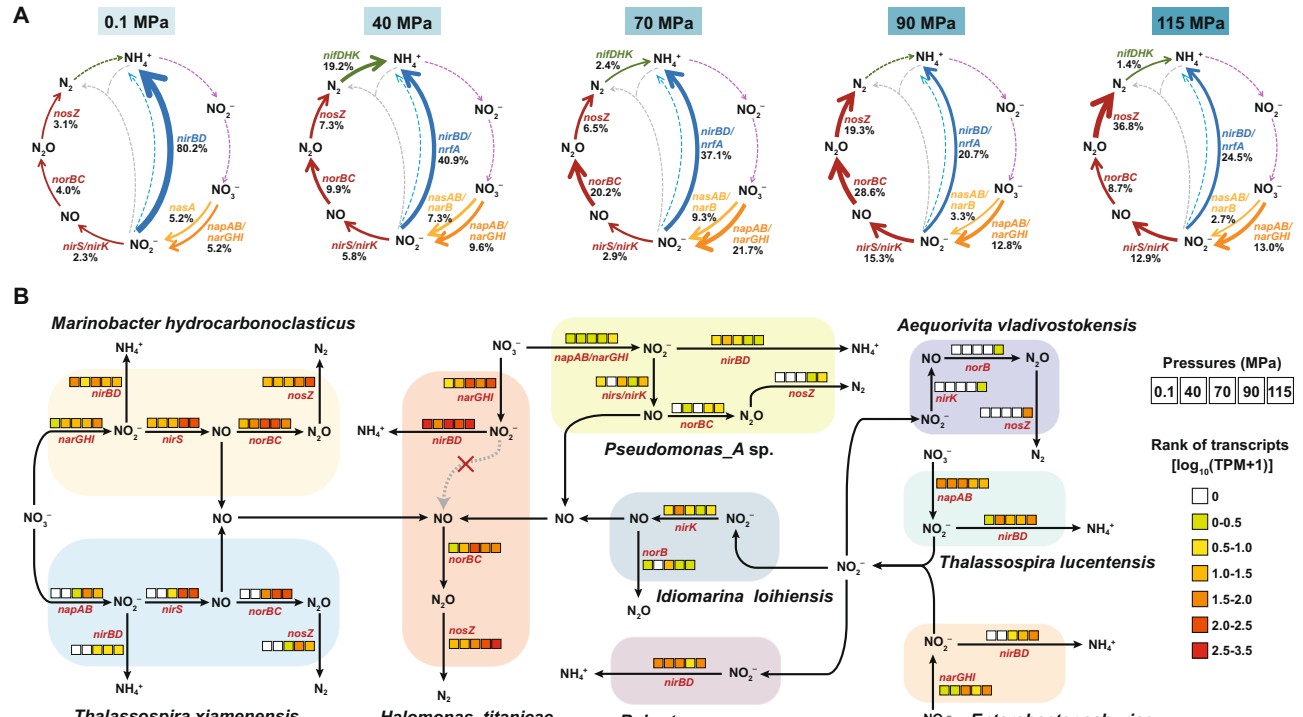

**Fig. 4 | Nitrogen-cycling genes transcriptional activity for continuous flowing incubation sediment samples under different hydrostatic pressures.**
**A** Nitrogen cycle schematics display the average abundance of nitrogen-cycling transcripts (based on TPM values) per hydrostatic pressure condition (relative to nitrogen-cycling pathways overall). The percentage of gene transcripts associated with each pathway component is shown in black font. Colored arrows represent pathways (orange = the first step of dissimilatory nitrate reduction, yellow = the first step of assimilatory nitrate reduction, red = denitrification, blue = dissimilatory nitrite reduction to ammonium (DNRA), light blue = assimilatory nitrite reduction to ammonium, green = nitrogen fixation, purple = nitrification, gray = anammox). The dashed lines indicate that genes associated with these pathways were not

identified in this incubation system. **B** Actively transcribed nitrogen metabolic pathways within MAGs reconstructed during each incubation pressure and their collaboration in nitrate reduction pathway. Different colors show the transcriptional activity (expressed as $\log_{10}(TPM + 1)$) of nitrogen metabolic genes under different hydrostatic pressures. *narGHI* membrane-bound dissimilatory nitrate reductase, *napAB* periplasmic dissimilatory nitrate reductase, *nirK* copper-containing nitrite reductase, *nirS* cytochrome $cd_1$-containing nitrite reductase, *norBC* nitric oxide reductase, *nosZ* nitrous oxide reductase, *nrfAH* cytochrome c nitrite reductase, *nirBD* NADH-dependent nitrite reductase, *nasAB* assimilatory nitrate reductase, *narB* ferredoxin-nitrate reductase, *nirA* ferredoxin-nitrite reductase, *nifDHK* nitrogenase, and *amoABC* ammonia monooxygenase.

## The active microbial groups involved in denitrification during incubation

A total of 63 metagenome-assembled-genomes (MAGs) with completeness >50% and contamination <10% were obtained from the six metagenomic datasets derived from the inoculum and flowing incubation sediment samples (Supplementary Data 7). These retrieved MAGs were taxonomically annotated with the GTDB-Tk tool. They were mainly affiliated with Proteobacteria (43 MAGs), Bacteroidota (9 MAGs), Actinobacteriota (4 MAGs), Firmicutes (2 MAGs) and Thermoproteota (2 MAGs) (Supplementary Data 7). The normalized abundance (RPKG) of each recovered MAG is shown in Supplementary Fig. 4 and Supplementary Data 7.

We then identified the microorganisms that can participate in the denitrification pathway at different hydrostatic pressures based on the presence of genes associated with denitrification. Among the 63 MAGs, 40 MAGs contain these genes (Supplementary Data 8). We further mapped the metatranscriptomic reads to these 40 MAGs, and found that 31 MAGs were actively involved in the denitrification process under at least one hydrostatic pressure condition (Supplementary Data 8). Furthermore, the results showed that different microbial groups were dominant drivers of denitrification under various hydrostatic pressures (Fig. 4B). Specifically, *Halomonas titanicae* was the most abundant denitrifier across the entire incubation period, and its denitrification-related genes (*narGHI*, *norBC*, and *nosZ*) were more actively transcribed under higher hydrostatic pressures, especially under 70, 90 and 115 MPa. *Marinobacter hydrocarbonoclasticus* was dominant at 40 MPa with its abundance decreased under increased

pressures, but the transcriptional level of its denitrification-related genes (*nirS*, *norBC*, and *nosZ*) increased under the elevated pressures. The abundance of *Thalassospira xiamenensis* increased as pressures increased, but its genes associated with denitrification (*napAB*, *nirS*, *norBC*, and *nosZ*) were only active under 70, 90, and 115 MPa. *Idiomarina loihiensis* also participated in denitrification, in which the *nirK* and *norB* genes were actively transcribed during incubation, however, its abundance decreased with the elevated pressures. Lastly, the abundance of *Aequorivita vladivostokensis* remained similar during incubation, but its denitrification-related genes (*nirK*, *norB*, and *nosZ*) were transcribed only under 115 MPa (Supplementary Data 7, 8).

## Other nitrogen transformation processes occurred in this flowing incubation system

The *nifDHK* genes involved in nitrogen fixation were detected in samples incubated at 40, 70, and 115 MPa, with the highest transcriptional activity being detected at 40 MPa (Fig. 4A and Supplementary Fig. 2). The *nasAB* and *narB* genes involved in assimilatory nitrate reduction to nitrite were also identified and were most actively transcribed at 70 MPa (Fig. 4A and Supplementary Fig. 2). The *nirA* gene involved in assimilatory nitrite reduction to ammonia was identified only in the metagenome of the inoculum sediment sample, but its RNA transcript was not identified during the incubation (Supplementary Data 4). Additionally, we did not identify any genes associated with anammox (ie., *hzs* and *hdh*) during the incubation (Supplementary Data 4). Ammonia monooxygenase gene (*amoABC*) and hydroxylamine dehydrogenase gene (*hao*) involved in the nitrification

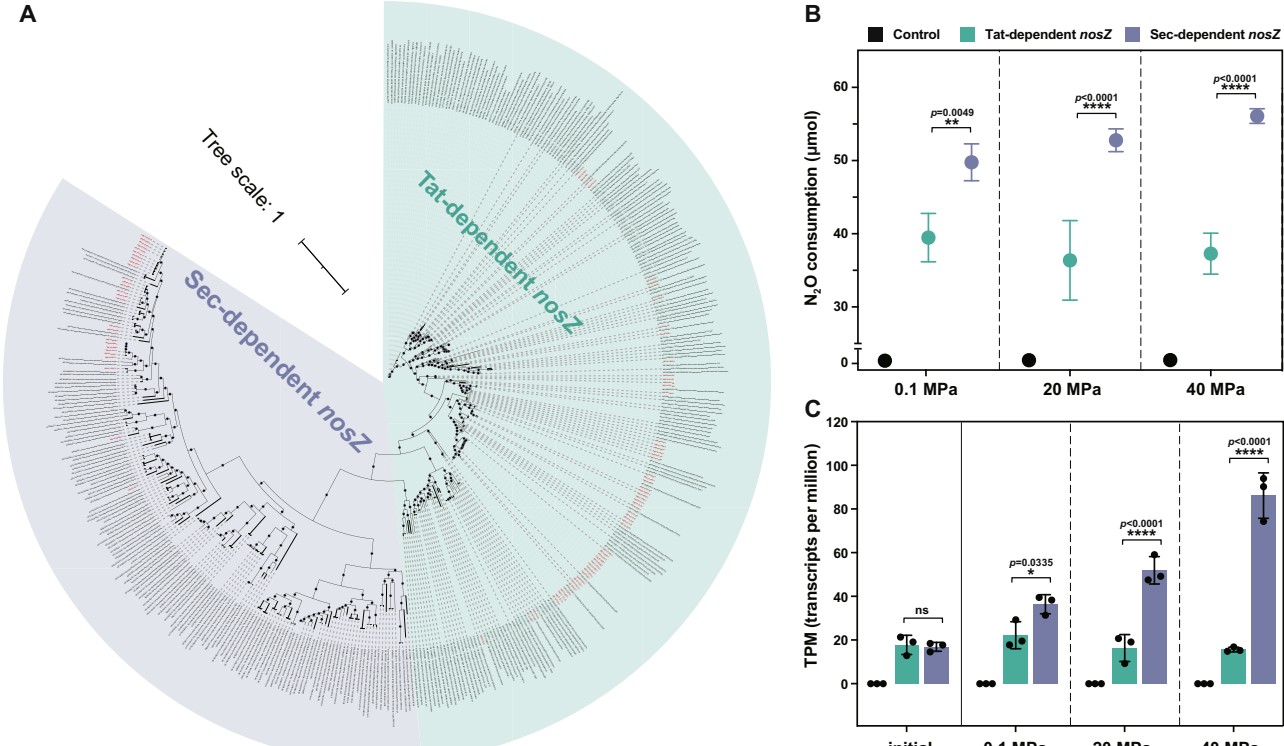

**Fig. 5 | The phylogenetic tree of NosZ protein sequences in this study with other known NosZ protein sequences from previous studies and the enzymatic evidence of N$_2$O reduction activity assay for two clades of nos gene cluster, Tat-dependent and Sec-dependent *nosZ* genes at hydrostatic pressures. A** The phylogenetic tree is constructed by IQ-TREE2 with the LG + F + R9 model. All *nosZ* gene sequences in this study are in red font. The Sec-dependent *nosZ* amino acid sequences are marked in purple and the Tat-dependent *nosZ* amino acid sequences are marked in green. **B** Monitoring of the change in N$_2$O consumption by Tat-dependent and Sec-dependent *nosZ* genes after incubation at 0.1, 20, and 40 MPa for 24 h. The N$_2$O consumption data are presented as mean values with standard deviation based on $n = 3$ biologically independent samples. **C** Barplot shows the transcriptional activity (TPM) of Tat-dependent and Sec-dependent *nosZ* genes after incubation at 0.1, 20, and 40 MPa for 24 h. The TPM data are presented as mean values with standard deviation based on $n = 3$ biologically independent samples. The $p$ values are analyzed using multiple *t*-tests. The significant variables are indicated by an overlying line and asterisk (ns, no significant; * $p < 0.05$; ** $p \leq 0.01$; *** $p \leq 0.001$; **** $p \leq 0.0001$). Source data are provided as a Source Data file.

pathway were not detected during the entire flowing incubation period (Supplementary Data 4). This seemed to contradict the dominance of ammonia-oxidizing archaea *Nitrosopumilaceae* in the archaeal community obtained by 16 S rRNA gene sequencing (Fig. 2B). However, considering that the cell number of archaea, which consistently decreased with the elevated hydrostatic pressures, was two orders of magnitude lower than that of bacteria, the absence of ammonia monooxygenase gene in subsequent metagenome and metatranscriptome may be explained by their low abundance and insufficient sequencing depth.

### The simultaneous denitrification and aerobic respiration under high hydrostatic pressure

The simultaneous denitrification and aerobic respiration in trench microorganisms under high hydrostatic pressure was verified in both in vitro and in situ surface sediment samples. In the flowing incubation experiments, the key genes (e.g., *Cyo*, *Cyd*, *Cco*, and *Cox*) involved in terminal oxygen reduction were abundant and actively transcribed under all incubation pressures (Supplementary Fig. 1), with the denitrification activity being enhanced with increased pressure (Fig. 4A). In addition, we further analyzed the transcription of genes associated with these pathways at the MAGs level. Our results showed that aerobic respiration-related genes encoding terminal oxidases and denitrification-related genes were actively transcribed concurrently in many taxonomic groups, such as *Halomonas titanicae*, *Marinobacter hydrocarbonoclasticus*, and *Thalassospira xiamenensis* (Supplementary Data 9).

Furthermore, to confirm this occurrence is happening in natural trench environments, we analyzed the metatranscriptomic dataset of in situ fixed surface sediments (0–10 cm) from the Mariana Trench. These sediment samples were immediately fixed with RNALater after being collected at the trench bottom (see "Methods"), therefore the metatranscriptomic analysis on these samples could reveal the in situ microbial activities. Our results showed that genes related to aerobic and anaerobic energy metabolic pathways, such as respiratory electron transport chains, denitrification, and TCA cycle were actively transcribed simultaneously (Fig. 6A and Supplementary Data 10). Additionally, these genes were mainly assigned to Proteobacteria, Actinobacteriota, Chloroflexota, Gemmatimonadota, Planctomycetota, and Thermoproteota (Fig. 6A). Thus, Proteobacteria and Actinobacteriota are concurrently capable of aerobic oxidation of organic matter and denitrification pathway, which is consistent with the flowing incubation experiments (Supplementary Data 9).

## Discussion

In this study, simultaneous aerobic respiration and denitrification were observed in both incubation experiments and in situ hadal trench environments, and increasing hydrostatic pressure promoted microbial denitrification activities. The simultaneous oxygen consumption and denitrification have been reported in permeable sediments from coastal ocean[21], but it is reported here for the first time that high hydrostatic pressure promotes denitrification even under oxic conditions. We propose that the enhanced denitrification at oxic conditions under high hydrostatic pressure could be the result of energy

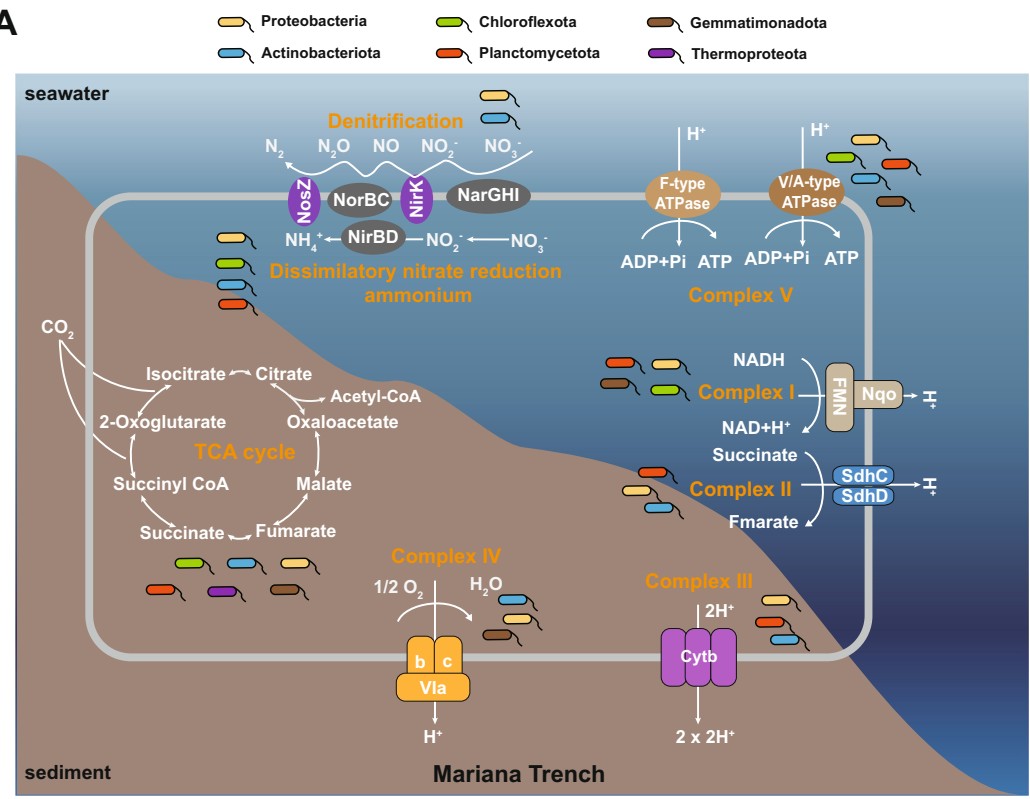

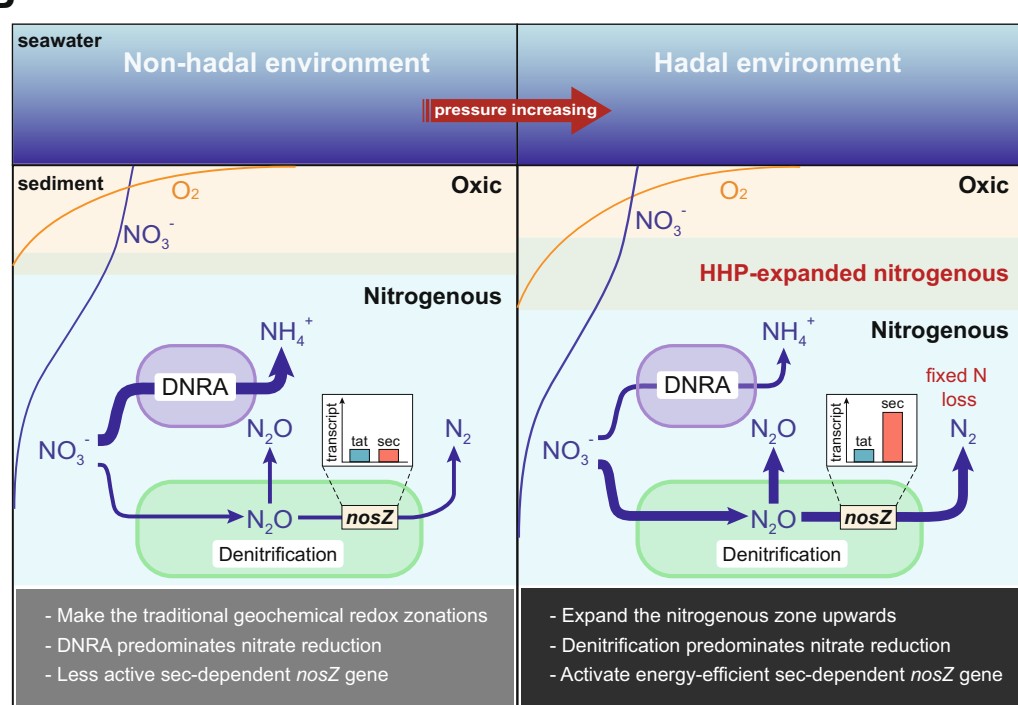

**Fig. 6 | The active microbial metabolism pathways of in situ RNA-fixed surface sediments in the Mariana Trench and a conceptual model that the mechanism of high hydrostatic pressure on nitrogen transformations in deep ocean sediments. A** Schematic representation of the active taxonomic groups involved in denitrification and aerobic respiration pathways in sediment from the Mariana Trench via metatranscriptomic analysis. The respiratory electron transport chain comprising complexes I to V, the TCA cycle, denitrification, and dissimilatory nitrate reduction to ammonium pathways are shown in orange font. **B** Proposed conceptual model of the effect of high hydrostatic pressures on redox zonations and nitrogen transformations in hadal trench sediments. High hydrostatic pressures stimulate the anaerobic nitrate-reducing activity under oxic conditions in hadal environments, where the main functional microorganisms coordinate and integrate energy-efficient tactics for survival, which further leads to a shallower depth of typical nitrogenous zone of chemical zonations in sediments. The dominant pathways are highlighted with thick arrows. The oxic sediment layer is painted pink, the nitrogenous sediment layer is painted blue, and the mixed layer in between is painted gray. The profiles of oxygen and nitrate are illustrated as orange lines and blue lines.

preservation requirements and to avoid oxidative stress response. By genome-centric analysis, our results showed that aerobic respiration-related genes and denitrification-related genes were transcribed concurrently under higher pressures within a diverse range of denitrifiers, such as *Halomonas titanicae*, *Marinobacter hydrocarbonoclasticus* and *Thalassospira xiamenensis* (Fig. 6A). This is consistent with previous studies that energy metabolism and high-pressure adaptation were coordinated in marine denitrifiers and nitrate reducers. For example, the transcriptional level of genes involved in energy metabolic pathways, such as glycolysis, dissimilatory nitrate reduction, denitrification, and TCA cycle was stimulated by high hydrostatic pressure in *Halomonas titanicae* ANRCS81[22]. Marine nitrate reducers, *Caminibacter mediatlanticus* and *Thermovibrio ammonificans* with higher-energies yielding capacity, were better adapted to high hydrostatic pressures[23]. Additionally, as shown in this study, the energy preservation mechanism was also observed within the denitrification pathway, where Sec-dependent $N_2O$ reductase dominated over Tat-dependent (Fig. 5). Similar energy conservation mechanism has also been reported in extreme environment-tolerant *Pyrobaculum calidifontidis* and *Salinibacter ruber*, both of which contain Sec-dependent $N_2O$ reductase[24]. The dominance of the Sec-dependent *nosZ* genes also occurs in low-nutrient groundwater[25]. Besides, high hydrostatic pressures induce intracellular oxidative stress (Supplementary Fig. 5), which impairs physiological functions[17,22,26,27]. Under oxidative stress, cells prefer to use nitrate rather than oxygen as the terminal electron acceptor to prevent further reactive oxygen species (ROS) accumulation[22]. Thus, energy preservation for cellular maintenance and the avoidance of oxidative stress under high hydrostatic pressures could be one of the mechanisms behind the enhancement of denitrification over aerobic respiration even under oxic conditions.

Chemical zonation (i.e., redox zone) refers to the vertical subdivision of natural environments based on the sequential thermodynamic availability of electron acceptors to oxidize organic matter during respiration processes[28,29]. The redox zones in the sedimentary environment were typically subdivided into aerobic respiration, nitrate reduction, manganese reduction, iron reduction, sulfate reduction, and methanogenesis zones[29]. The enhanced denitrification at high hydrostatic pressure under oxic conditions potentially leads to the restructuring of chemical zonation in deep-sea surface sediments and subsequently revises geochemical cycling processes therein. The enhanced denitrification at oxygenic niches, where denitrification was previously considered impossible, will lead to an upward increased thickness of the typical nitrogenous zone in sediments, and subsequently, the enhanced nitrogen loss (Fig. 6B). The enhanced denitrification under high hydrostatic pressure and its co-existence with aerobic respiration has not been recognized in the geochemical gradients and models[30,31]. The acknowledgment of such a phenomenon could greatly broaden our understanding of the biogeochemical cycling of key elements in hadal trenches and give rise to new research frontiers.

Denitrification and anammox are the main pathways leading the fixed nitrogen loss in trenches, but with distinct environmental niche preferences[14,32-34]. The distribution of anammox bacteria varied substantially in the global ocean water column, but preferred oxygen deficient zone[35]. The maximum $N_2$ production rates were reported to be about 0.02 - 2 nmol N/(day·cm³) in the Kermadec Trench and Atacama Trench, where both ecosystems were dominated by anammox bacteria[8]. In comparison, denitrifiers are widely distributed in the global ocean, especially abundant in the sediment with high organic matter[36]. We demonstrated that denitrification was responsible for the continuous nitrogen loss when organic carbon was supplied (Figs. 3 and 4A, "Methods"). Similar results have also been reported in the Yap Trench[16] and Challenger Deep sediments[37]. Under laboratory conditions, we observed that the rate of nitrogen loss (N−$N_2O$) caused by denitrification-derived $N_2O$ alone was ~ 4.6−nmol

N/day/mL under 70 MPa with 125 mL of Mariana Trench sediment as the initial inoculum (see "Methods") Although the $N_2$ production was unfortunately not measured because of technical limitation, the denitrification-derived $N_2$ is expected to contribute to additional nitrogen loss. Moreover, complete denitrification often requires a complex microbial consortium to achieve[38], a series of partial products ($NO_2^-$, NO, $N_2O$) may be generated, which allows syntrophic relationships between denitrifiers and other microorganisms to be established[7,9,37]. This suggests that the denitrification-dominated ecosystem could support a greater microbial diversity than the anammox-dominated ecosystem, thus providing a possible mechanism for the maintenance of microbial diversity in hadal trenches.

Here we demonstrated that denitrification is the bio-preferable energetic pathway under high hydrostatic pressures and contributes to the nitrogen loss from generally oxygenated upper layer sediment in hadal trenches. We observed that high hydrostatic pressure promotes denitrification activity even under oxic conditions, presenting an indication that hydrostatic pressure has the potential to modify the niche breadth and the activity of microorganisms. This modification, in turn, influences their contributions to the elemental cycling processes within hadal trenches. The differential distributions of denitrifiers and anammox bacteria with varied nitrogen loss rates suggest that nitrogen loss hotspots may exist across the global ocean floor.

## Methods

### Samples collection and continuous flowing incubation at different hydrostatic pressures under oxic conditions

The sediment sample for high-pressure incubation was obtained by the Jiao Long Human Occupied Vehicle (HOV) from the northern slope of the Mariana Trench (142.2516°E, 11.6639°N) at Dive 119 station with a water depth of 6002 m during the cruise on R/V Xiangyanghong09 Cruise DY37-II in June 2016. The collected sediments were sliced into 4 cm interval layers and were immediately preserved in multiple 70 MPa (approximately the pressure at the sampling site) pressure-retaining vessels on shipboard and stored at 4 °C until used for subsequent experiments. In addition, we collected surface sediment at the Mariana Trench at water depths of 8207–10,898 m during the R/V Tansuoyihao Cruise TS-21 from August to December 2021 (142.5947°E, 11.3649°N; 142.5926°E, 11.3867N; 142.5869°E, 11.3740 °N; 142.5602°E, 11.3619°N; 142.5602°E, 11.3619°N; 142.1562°E, 11.1615°N; 142.1572°E, 11.1590°N; 142.3429°E, 11.1970°N; 142.3429°E, 11.1970°N; 142.2038°E, 11.3393°N; 142.2166°E, 11.3350°N). These sediments were collected and immediately fixed with RNALater at in situ condition via a specially designed in situ fixation sampler, and used for metatranscriptomic analysis. Sample collection and transportation have been permitted by the Federated States of Micronesia, with the permit number FM-XXRS-23522. The research complies with all relevant ethical regulations.

The above 20 cm layers of sediments were used as the inoculum for the incubation (hereafter referred to as inoculum sediments) (Fig. 1A). The concentration of $NO_x^-$ ($NO_2^-$ and $NO_3^-$) in the overlying water sample was 213.7 μg/L (1.98 μmol/L) as shown in the previous study[39]. During incubation, the inoculum sediment was suspended in the chemically defined marine medium which was modified from Widdel and Bak[40] and contained (/L): NaCl (Cat. No. 10019318, Sinopharm Chemical Reagent, China) 26.0 g, $MgCl_2·6H_2O$ (Cat. No. 10012818, Sinopharm Chemical Reagent, China) 5.0 g, $CaCl_2·2H_2O$ (Cat. No. 20011160, Sinopharm Chemical Reagent, China) 1.4 g, $Na_2SO_4$ (Cat. No. 10020518, Sinopharm Chemical Reagent, China) 4.0 g, $KH_2PO_4$ (Cat. No. 10017618, Sinopharm Chemical Reagent, China) 0.1 g, KCl (Cat. No. 10016318, Sinopharm Chemical Reagent, China) 0.5 g, D-( + )-glucose (Cat. No. DX0145, Sigma-Aldrich, Germany) 5.4 g, $NaNO_3$ (Cat. No. 10019918, Sinopharm Chemical Reagent, China) 100 μmol (as sole nitrogen source), bicarbonate solution 30 mL, trace element mixtures 1 mL, vitamin mixture 1 mL, thiamine solution 1 mL, and vitamin $B_{12}$ solution 1 mL. Glucose was provided as the organic

carbon source and electron acceptor to investigate the microbial nitrate reduction process, according to previous studies[41,42]. The homogenized slurry was purged with continuous argon flushing to achieve a dissolved oxygen concentration of 100 μmol/L (this value is based on the oxygen concentration of ~100 μmol/L in the 0–20 cm sediment layers at ~6000 m deep site[3]). Then, it was transferred into the Deep Ocean Experimental Simulator (DOES, which is specially designed to construct an environment in the laboratory that is similar to the deep biosphere, in terms of temperature, hydrostatic pressure, flow rate, pH, nutrient availability, etc.[19]) and incubated in a flow-through mode at a flow rate of 0.3 mL/min (Fig. 1B). The incubation was performed at 4 °C, and the incubation hydrostatic pressures (0.1 MPa, 40 MPa, 70 MPa, 90 MPa and 115 MPa) were changed every 15 days.

### Biochemical analysis

To monitor the metabolic activities, the consumption of nitrate, as well as the production of ammonium, nitrite, and nitrous oxide were analyzed. Sampling of the above chemical parameters was conducted every 3 days without depressurization throughout the incubation experiments. The concentrations of dissolved inorganic nitrogen species ($NO_3^-$, $NO_2^-$, and $NH_4^+$) were quantified using an AA3 Auto-Analyzer system (Seal Analytical, UK). The concentration of gas $N_2O$ was measured by an Agilent 6890 N Gas Chromatograph (Agilent Technologies, USA). The initial concentration of dissolved oxygen was measured using the Microx 4 oxygen meter (PreSens, Germany). The concentration difference between the inlet and outlet of this DOES system was multiplied by the flow rate (0.3 mL/min) to calculate the corresponding consumption and production rates of these nitrogenous compounds (nitrate and nitrite, ammonium, nitrous oxide). We further estimated the nitrogen loss rate (N–$N_2O$, nmol N/day/mL) due to denitrification-derived $N_2O$ alone under 70 MPa using 125 mL of Mariana Trench sediment as the inoculum. The corresponding calculation formula is as follows: net nitrous oxide production rate*2*1000/125 mL.

### Nucleic acid extraction and metagenomic/metatranscriptomic sequencing

Biomass sampling was performed when switching incubation pressure (Fig. 1C). Total microbial genomic DNA was extracted and purified using the modified SDS-based method described by Natarajan et al.[43], and stored at −20 °C before further assessment. The quantity and quality of extracted DNA were measured using Qubit 4.0 Fluorometer (Invitrogen, Carlsbad, CA, USA) and agarose gel electrophoresis, respectively. The extracted microbial DNA was processed to construct metagenome shotgun sequencing libraries with insert sizes of 350 bp following the standard Illumina TruSeq DNA Sample Preparation Guide. Each library was sequenced by Illumina NovaSeq 6000 platform (Illumina, USA) with PE150 strategy at Shanghai Personal Biotechnology (Shanghai, China). The extraction of RNA from sediment samples was carried out using the RNeasy® PowerSoil® Total RNA Kit (Cat. No. 12866-25, Qiagen, Germany) according to the manufacturer's instructions, then quantified using a Qubit 4.0 Fluorometer (Invitrogen, Carlsbad, CA, USA). To ensure DNA removal, the RNA extracts were treated with TURBO DNase (Cat. No. AM2238, Invitrogen, Waltham, MA, USA) as directed by the manufacturer. The purified RNA was converted to cDNA, then the metatranscriptomic library was constructed by using Illumina TruSeq Stranded mRNA LT Sample Prep Kit, and subsequent sequencing as described above.

### 16 S rRNA gene sequencing and quantitative PCR (qPCR) analysis

The V4–V5 region and V4 region of archaeal and bacterial 16 S rRNA genes were amplified by polymerase chain reaction (PCR) with the primer pair Arch516F (5′-TGYCAGCCGCCGCGGTAAHACCVGC-3′)/

Arch855R (5′-TCCCCCGCCAATTCCTTTAA-3′), and the primer pair Bac533F (5′-TGCCAGCAGCCGCGGTAA-3′)/Bac806R (5′-GGACTAC-CAGGGTATCTAATCCTGTT-3′), respectively. The PCR amplification procedure was previously described[44]. Subsequently, the PCR products were purified using an EZNA Gel Extraction Kit (Cat. No. D2500-01, Omega Bio-Tek, Norcross, GA, USA). The purified DNA was sequenced on the Illumina NovaSeq platform by Shanghai Personal Biotechnology (Shanghai, China). Microbiome analysis was performed with QIIME2 (v.2020.8)[45]. The paired-end raw sequence data containing the forward or reverse reads for each sample were demultiplexed and quality-filtered using the q2-demux plugin. The paired-end demultiplexed sequences were denoised by DADA2[46] (via q2-dada2). In DADA2, the sequence bases with quality score greater than 20 (i.-e., < 1% error rate) were kept based on the resulting interactive quality plot of the previous step, and these command parameters (i.e., --p-trunc-len-f, --p-trunc-len-r, --p-trim-left-f, --p-trim-left-r, --p-chimera-method) were applied to truncate and trim the read sequences and remove chimeras. The taxonomy classification and taxonomic analysis were assigned to the ASVs by comparing the query sequences to a reference SILVA 138 database using the q2-feature-classifier[47] classify-sklearn Naive Bayes taxonomy classifier.

The quantification of archaeal and bacterial 16 S rRNA genes was respectively conducted using the primer pair Bac341F (5′-CCTACGGGWGGCWGCA-3′)/Bac519R (5′-TTACCGCGGCKGCTG-3′), and the primer pair Uni519F (5′-GCMGCCGCGGTAA-3′)/Arch908R (5′-CCCGCCAATTCCTTTAAGTT-3′). The 7500 Real-Time PCR System and Power-Up™ SYBR™ Green Master Mix (2X) (Cat. No. A25741, Applied Biosystems, Foster City, CA, USA) were used for all qPCR analysis according to the manufacturer's instructions. The archaeal and bacterial cell counts were calculated using archaeal and bacterial 16 S rRNA gene copy numbers divided by 1.7 and 5.2 (mean 16 S rRNA operon copy number)[48], respectively.

### Metagenomic and metatranscriptomic analysis

The omics analysis was performed on both gene-centric strategy and genome-centric strategy. Briefly, the 150 bp paired-end raw reads were first trimmed by BBDuk tool (v.38.96) (https://sourceforge.net/projects/bbmap/) with a sequence quality score of > 20 and a final minimum length of > 90 bp. Obtained clean reads were assembled by SPAdes (v.3.12.0)[49] with "--meta --only assembler -k 65,75,96,115,127". The assembly was filtered for a minimum length of 500 bp using a custom Python script[50]. All metatranscriptomic reads were first filtered by BBDuk tool (v.38.96) and then aligned to a combined rRNA database from SILVA and Rfam[51] using Bowtie2 (v.2.4.1)[52]. The unaligned mRNA reads were collected for quantification of gene expression.

For the assembled contigs, genes were predicted by Prodigal (v.2.6.3)[53] for the filtered assembly and those with lengths smaller than 100 bp were discarded. The modified gene set was functionally annotated with an integrated result, with the following priorities: Ghost-KOALA (v.2.2)[54] > emapper (v.2.0.1) against the EggNOG database (v.5.0)[55,56] > KofamKOALA (v.1.0.3)[57]. Clean metagenomic and metatranscriptomic reads from each incubation pressure were mapped to the assembly by BBMap (v.38.24) with "k = 13 minid = 0.95 pairlen = 350 resccuedist = 650". The mapped file in SAM format was converted to BAM format and sorted by SAMtools (v.1.15.1)[58]. For metagenomic datasets, the depth of each scaffold in every incubation was determined by the script "jgi_summarize_bam_contig_depths" from MetaBAT2 (v.2.15)[59] with the default parameters. FeatureCounts (v.1.5.3)[60] was used to count the read number of each gene, and the transcripts per million (TPM) value was calculated with a custom Python script[50].

Three binning software programs were used to obtain the primary MAGs, as described in the previous study[61]. For MetaBAT2 (v.2.15), different sensitivities (--maxP 60, 75, and 90) and specificities (--minS 60, 75, and 90) were combined. The two marker gene sets (40 and 107)

were analyzed by MaxBin (v.2.2.6)[62]. CONCOCT (v.1.0.0) analysis[63] was also carried out. Then DAStools (v.1.2.2)[64] was used to integrate the results to calculate an optimized, nonredundant set of MAGs for each incubation. Quality and taxonomy were determined by CheckM (v.1.1.5)[65] and GTDB-Tk tools (v.2.3.2)[66] with the GTDB r202[67] respectively. MAGs with completeness >50% and contamination <10% were dereplicated by dRep (v.3.4.2)[68] with 95% average nucleotide identity (ANI) to obtain a non-redundant MAG set for the whole incubations. Clean metagenomic and metatranscriptomic reads from each incubation pressure were mapped to the MAG set for gene quantification with the methods described above. The normalized abundance of recovered MAGs was evaluated by the RPKG (reads recruited per kilobase of genome per gigabase of metagenome) values for the comparison among different genomes and metagenomes, as described by Liu et al.[69].

### Heterologous expression and activity assay of $N_2O$ reductase genes

The phylogenetic tree of NosZ amino-acid sequences was constructed together with the reference sequences from a previous study[70] by IQ-TREE tool (v.2.1.2)[71] with LG + F + R9 model using 1000 ultrafast bootstrap replicates and 1000 bootstrap replicates for SH-aLRT. The *nosZ* gene clusters, including Tat-dependent (clade I) and Sec-dependent (clade II) clades belonging to Halomonadaceae and Flavobacteriaceae families respectively, were synthesized by Shanghai Saiheng Biotechnology Co., Ltd (Shanghai, China). The two gene clusters were subsequently cloned into the pSW2 expression vector between the sites BamHI and SalI[72]. These three plasmids (pSW2-Tat, pSW2-Sec and pSW2) were heterologously expressed in deep-sea model bacterium *Shewanella piezotolerans* WP3NR (a piezotolerant strain with the pressure range of 0.1–50 MPa)[73] by conjugal transfer using *E. coli* WM3064 strain (Fig. 1D). Three transconjugants (WP3NR-pSW2-Tat, WP3NR-pSW2-Sec and WP3NR-pSW2) were selected by chloramphenicol resistance and were verified via colony PCR for subsequent experiments. The obtained strains were cultured in modified marine 2216E medium (5 g/L tryptone (Cat. No. LP0042, Thermo Scientific™ Oxoid™, USA), 1 g/L yeast extract (Cat. No. LP0021, Thermo Scientific™ Oxoid™, USA), 34 g/L NaCl (Cat. No. 10019318, Sinopharm Chemical Reagent, China)) to activate cells. At the early stationary phase ($OD_{600} = 1$), cells were washed with culture salt solutions to remove the residual organic matter and then transferred into the defined LMO-812 minimal medium[74]. The inoculum was then purged with argon gas to ensure anoxic conditions. To determine $N_2O$ reduction activity, 15 mL inoculum was amended with 500 μmol acetate and 2 mL (-83 μmol) of $N_2O$ gas (99.999%), then was incubated under 0.1, 20, and 40 MPa for 24 h at 15 °C. The $N_2O$ concentrations were determined by an Aligent 6890 N gas chromatograph (Agilent Technologies, USA). After the incubation, total RNA was extracted with the common TRIzol method[27], and RNA samples in triplicate were subjected to quality control and sequencing on the Illumina platform at Shanghai Personal Biotechnology (Shanghai, China). The subsequent transcriptomic analysis was performed as described previously[22].

### Reporting summary

Further information on research design is available in the Nature Portfolio Reporting Summary linked to this article.

## Data availability

The raw 16 S rRNA gene amplicon reads metagenomic and metatranscriptomic data of flowing incubation samples, the transcriptomic data of recombinant WP3NR strains, and the metatranscriptomic data of in situ fixed sediments in the Mariana Trench generated in this study have been deposited to the National Omics Data Encyclopedia (NODE, https://www.biosino.org/node/index) database under the accession numbers OEP004015, OEP004042, OEP004045, OEP004102, OEP004512, as well as the NCBI SRA database under the BioProject IDs of PRJNA1083314, PRJNA1083644, PRJNA1083643, PRJNA1083642. The databases used in this study include SILVA 138 database (https://www.arb-silva.de/documentation/release-138/), GTDB database Release 202, and KEGG database (https://www.genome.ad.jp/kegg/). All other data are available in this paper or the Supplementary Information. Source data are provided in this paper.

## Code availability

The custom Python scripts used in this study are available in the Figshare database (https://doi.org/10.6084/m9.figshare.25331617). Software versions and non-default parameters used in this paper have been appropriately specified where required.

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

## Acknowledgements

The authors acknowledge the crew of R/V Xiangyanghong09 Cruise DY37-II and Tansuoyihao Cruise TS-21. We thank the pilots of Fendouzhe HOV and Jiaolong HOV, Prof. Shanya Cai, Xiang Xiao, and Yinzhao Wang for their help with the sample collection. The authors thank Dr. Jialin Hou and Qilian Fan for their kind help in figure drawing and text checking. This work was supported by the National Natural Science Foundation of China (Grant No.: 92251303, 42122043, 42188102, 91951117), Shanghai Pilot Program for Basic Research-Shanghai Jiao Tong University (No.: 21TQ1400201), and National Key Research and Development Program of China (Grant No.: 2023YFC2812800).

## Author contributions

N.Y and Y.Z. designed the research; N.Y. and Y.L. performed the experiments and conducted the data analysis; S.W. sampled the sediment from Mariana Trench; N.Y., Y.L., M.J., S.W., and Y.Z. wrote the paper.

## Competing interests

The authors declare no competing interests.
