## [Peer Review File · Nature Communications]

High hydrostatic pressure stimulates microbial nitrate reduction in hadal trench sediments under oxic conditionsREVIEWERS' COMMENTS:

Reviewer #1 (Remarks to the Author):

Yang et. al collected sediment samples from hadal zone of Mariana Trench and incubated in a chemostat with increasing pressure condition to explore the denitrification process mediated by hadal microbial communities. This study presents one of the few datasets that have examined nitrogen cycles in deep trenches, and the combination of omics tools, biochemistry measurement and microbiology experiment provide new insights on how high hydrostatic pressure (HHP) will induce denitrification in the trench sediments. While the paper does report novel results, I found that the authors did not successfully convey the clarity of their results. The authors can add more details to this manuscript and fully explain their findings if the article type allows.

Specific comments:

L69-75: the sediment sample were collected at 6000m but pre-incubated under 70MPa before transferred to the chemostat, it is no longer 'in-situ' condition.

L74-75: 'better growth' better than which condition? Or does here mean, in the chemostat the microbes grew better under 70MPa than the pre-incubation (which is also 70MPa)? This brings out another question, before transferring to the chemostat, there were de-pressurizing process and nutrient supplement, drastic changes were found in taxonomic compositions, cell abundances and N compounds consumption/production (Fig 2-3), how would this pre-processing affect the conclusion that sediment microbes grew 'better' under 70 MPa?

L99-110, L112-134 here the omics data were used to support the idea that HHP drove the denitrification of sediment microbial community even under oxic conditions. The conclusion heavily relies on the transcriptis abundance of targeted genes (nar, nap, nir and nos) but the taxonomic affiliation of those gene/transcripts are missing. I can get some clue from Extended data fig 2 but the direct link is missing. For example, the gene abundance of nirB and nosZ are relatively stable in the cheomstats despite pressure changes, is it related to Halomonadaceae, which also didn't change so much? This is important because the community shifted from pre-incubation to chemostat incubation, one may wonder how the total community(16S, metagenome) and active community (metatranscriptome) will change accordingly.

Also I can see ammonia oxidation in the pre-incubation sample, but the gene/transcripts were completely gone after transfer into chemostat(Fig 4, table S4), how do the authors explain this data with the dominance of Nitrosopumilaceae from all conditions? Does this mean the amoABC are from AOB not AOA? I would suggest the author carry out in-depth analysis with the omics dataset to pinpoint the key species responding to HHP induced denitrification and discuss the differences between pre-incubation and chemostat conditions. MAG analysis can provide species level resolution especially for the chemostat based experiments and both metagenome and metatranscriptome will help identify the physiologic responses to HHP.

L137-158: The data here nicely showed HHP can induce the expression and functioning of nosZ. As far as I can see, the metagenome were assembled separately, with 59 TAT-nosZ and 27 Sec nosZ, what is the similarity for each group of nosZ, are they redundant (both occurrence and gene abundance) when the pressure changes? Nevertheless, this question is minor, the recombinant experiment provides direct evidence for the nosZ activity under HHP. Fig.5A is hard to read, I would suggest separate the transcripts abundance data from the tree.

L173: O₂ consumption data is missing. Or is this sentence related to Extended fig1?

L180-181: this sentence is misleading. Does this mean the denitrifiers can also use O₂ or denitrifier can live under oxic conditions while others are using O₂?

L222: 30.3 nmol N /day cm³. How is this value calculated? From table S3, the N₂O production rate is 0.287 umol/day under 70MPa, $0.287 \times 2 \times 1000 / 30.3 = 18 \text{ cm}^3(\text{ml})$, $18 \text{ ml} / 0.3 \text{ ml}/\text{min} = 60 \text{ min}$ (1H), I hope my calculation is wrong, otherwise the N₂ production is equivalent to Atacama Trench.

L 247-249: contextual data for sampling site is vague. What is the NO₃ concentration of ambient water, how deep was the sediment sample collected, top 5cm?

Overall I would suggest the authors focus more on the physiological response of denitrifiers to HHP rather than getting back to 'in-situ' biogeochemical mechanisms as the pre- and after- incubations drives too far away from real scenarios.

Reviewer #2 (Remarks to the Author):

The manuscript entitled "High hydrostatic pressure stimulates microbial nitrate reduction in

hadal trench sediments under oxic condition” by Yang et al. investigated the microbiology and omics of a marine sediment sample collected at a depth of over 6,000 m. This work demonstrated that denitrification is a major metabolic process in the nitrogen cycle under aerobic conditions and high hydrostatic pressure. The results are interesting, well presented and documented. However, this work lacks control experiments (e.g. in the absence of oxygen) in my view, and the experimental design is inadequate, as samples pressurized and maintained at in situ hydrostatic pressure during collection and storage, were directly incubated first at atmospheric pressure and then successively at increasing pressures applied during incubation. The enrichment of microorganisms at atmospheric pressure undoubtedly led to growth and an increase in the ratio of non-piezophilic or piezotolerant microorganisms, which became dominant during incubation compared with microorganisms adapted to in situ pressure conditions. Metagenomic and metatranscriptomic analysis lacks analysis of gene diversity and binning in relation to the dominant taxonomic groups present. These results do not support the authors' conclusions regarding active and major metabolisms in situ.

Specific comments:

- Lines 86-88: Under your experimental conditions, can you tell us what role Thaumarchaea play in the nitrogen cycle?
- Lines 112-125: Could you indicate to which taxonomic group the genes described in this paragraph can be assigned?
- Line 142: Similarly, to which taxonomic group does the selected *nosZ* gene refer?
- Lines 180-181: this conclusion is not supported by the results obtained, i.e. what is the diversity of respiratory chains present and active, for example?
- Lines 247—249: What are the physicochemical characteristics of the sediment sample used in this study?
- Line 254: Why was glucose used as a source of carbon and energy?

Response to the reviewers' comments:

Reviewer #1 (Remarks to the Author):

Yang et. al collected sediment samples from hadal zone of Mariana Trench and incubated in a chemostat with increasing pressure condition to explore the denitrification process mediated by hadal microbial communities. This study presents one of the few datasets that have examined nitrogen cycles in deep trenches, and the combination of omics tools, biochemistry measurement and microbiology experiment provide new insights on how high hydrostatic pressure (HHP) will induce denitrification in the trench sediments. While the paper does report novel results, I found that the authors did not successfully convey the clarity of their results. The authors can add more details to this manuscript and fully explain their findings if the article type allows.

Response: We are glad to see that the reviewer appreciates the novelty of our research. The manuscript has been modified according to the reviewers' comments. Briefly, we have included a detailed description and discussion on the taxonomy of denitrifiers to clarify the key players. Moreover, we have added the metatranscriptomic data of the in situ fixed samples to demonstrate what we found in the lab also occurs in natural conditions.

Specific comments:

L69-75: the sediment sample were collected at 6000m but pre-incubated under 70Mpa before transferred to the chemostat, it is no longer 'in-situ' condition.

Response: The sediments were collected at 6002 m in the Mariana Trench and immediately preserved at 70 MPa (approximately the pressure at the sampling site) using a pressure-retaining vessel. We agreed that it was no longer an 'in-situ' condition, as the microbial community would have changed during the storage. Thus, we deleted these inappropriate descriptions and the modified contents as follows:

To investigate the influence of hydrostatic pressure on the microbial community and functions, the sediment sample, collected at a water depth of 6002 m in the Mariana Trench, was sequentially incubated at 0.1 MPa, 40 MPa, 70 MPa, 90 MPa, and 115 MPa for 15 days each with a continuous supply of nitrate and dissolved oxygen. With the elevated hydrostatic pressures, the cell numbers of both bacteria and archaea declined

(except for a slight increase for bacteria at 70 MPa). Nevertheless, the bacterial population consistently outnumbered the archaeal population by two orders of magnitude (Fig. 2A, B; Supplementary Table 1).

These modified contents were in the revised manuscript L68-75.

L74-75: 'better growth' better than which condition? Or does here mean, in the chemostat the microbes grew better under 70MPa than the pre-incubation (which is also 70MPa)? This brings out another question, before transferring to the chemostat, there were de-pressurizing process and nutrient supplement, drastic changes were found in taxonomic compositions, cell abundances and N compounds consumption/production (Fig 2-3), how would this pre-processing affect the conclusion that sediment microbes grew 'better' under 70 MPa?

Response: What we meant was that the microbes exhibited better growth than they were at the lower hydrostatic pressures (i.e., 0.1 MPa, 40 MPa). However, as the difference was not significant, we agree that this statement is not accurate. We have rephrased the sentence as "With the elevated hydrostatic pressures, the cell numbers of both bacteria and archaea declined (except for a slight increase for bacteria at 70 MPa)." in L71-73.

Regarding the reviewer's second concern, we agree that the pre-processing would affect the microbes. Therefore, we have only focused on the effect of pressure on the inoculum after being transferred into the flowing incubation system in the revised manuscript.

L99-110, L112-134 here the omics data were used to support the idea that HHP drove the denitrification of sediment microbial community even under oxic conditions. The conclusion heavily relies on the transcript's abundance of targeted genes (nar, nap, nir and nos) but the taxonomic affiliation of those gene/transcripts are missing. I can get some clue from Extended data fig 2 but the direct link is missing. For example, the gene abundance of nirB and nosZ are relatively stable in the chemostats despite pressure changes, is it related to Halomonadaceae, which also didn't change so much? This is important because the community shifted from pre-incubation to chemostat incubation, one may wonder how the total community (16S, metagenome) and active community (metatranscriptome) will change accordingly.

Response: We have added the taxonomic affiliation of these actively transcribed genes at the MAGs level. In brief, we obtained 40 MAGs that contain nitrogen-cycling associated genes. Additionally, these nitrogen-cycling associated genes were actively transcribed in 31 MAGs. These 31 MAGs were mainly affiliated with Proteobacteria (28 MAGs), Bacteroidota (8 MAGs), Actinobacteriota (3 MAGs) and Thermoproteota (1 MAG).

Regarding the *nosZ* and *nirB* genes, most of them were identified from *Halomonas titanicae* (Halomonadaceae), which remained dominant under every incubation pressure. We have added a new descriptive paragraph (L167-196), new figures (Fig. 4B and Extended Data Fig. 4), added Supplementary Tables 7 and 8 in the revised manuscript. These results are also discussed in the revised manuscript L251-255.

Also, I can see ammonia oxidation in the pre-incubation sample, but the gene/transcripts were completely gone after transfer into chemostat (Fig 4, table S4), how do the authors explain this data with the dominance of Nitrosopumilaceae from all conditions? Does this mean the *amoABC* are from AOB not AOA?

Response: The *amoABC* genes in the inoculum sediment were affiliated with both AOB (Nitrosomonadaceae) and AOA (Nitrosopumilaceae). The disappearance of the *amoABC* genes may be because the sequencing depth was not high enough to identify low-abundance genes. Our results showed that the relative abundance of AOB (Nitrosomonadaceae) was extremely low (~0.03%) based on the amplicon sequencing results; whereas the AOA (Nitrosopumilaceae) was two orders of magnitude lower than bacteria (Fig. 2B). Therefore, the ammonia oxidizers are present throughout the incubation but was not detected in the metagenome. More sensitive methods such as amplicon sequencing targeting the *amoABC* genes may be required for the detection. Nevertheless, as these ammonia oxidizers are consistently present, ammonia oxidation activity is expected to present, but may be at a very low level given the low abundance of AOA and AOB.

The corresponding content was modified in the revised manuscript L207-216.

I would suggest the author carry out in-depth analysis with the omics dataset to pinpoint the key species responding to HHP induced denitrification and discuss the differences

between pre-incubation and chemostat conditions. MAG analysis can provide species level resolution especially for the chemostat based experiments and both metagenome and metatranscriptome will help identify the physiologic responses to HHP.

Response: We carried out additional genome-centric approaches at the MAGs level. We found that the taxonomy of dominant microbial groups driving denitrification changed under various hydrostatic pressures (Fig. 4B). Specifically, *Marinobacter hydrocarbonoclasticus* was dominant at 40 MPa with its abundance decreased under increased pressures, but the transcriptional level of its denitrification-related genes (*nirS*, *norBC*, and *nosZ*) increased under the elevated pressures. The abundance of *Thalassospira xiamenensis* increased as pressures increased, but its genes associated with denitrification (*napAB*, *nirS*, *norBC*, and *nosZ*) were only detected as being active under 70, 90, and 115 MPa. *Idiomarina loihiensis* also participated in denitrification, in which the *nirK* and *norB* genes were actively transcribed during incubation, however, its abundance decreased with the elevated pressures. The abundance of *Aequorivita vladivostokensis* remained stable during incubation, but its denitrification-related genes (*nirK*, *norB*, and *nosZ*) were transcribed only under 115 MPa. Lastly, *Halomonas titanicae* was the most abundant denitrifier across the entire incubation period, and its denitrification-related genes (*narGHI*, *norBC*, and *nosZ*) were more actively transcribed under high hydrostatic pressures, especially under 70, 90 and 115 MPa.

We have added a new descriptive paragraph (L167-196), new figures (Fig. 4B and Extended Data Fig. 4), added Supplementary Tables 7 and 8 in the revised manuscript.

L137-158: The data here nicely showed HHP can induce the expression and functioning of *nosZ*. As far as I can see, the metagenome were assembled separately, with 59 TAT-*nosZ* and 27 Sec *nosZ*, what is the similarity for each group of *nosZ*, are they redundant (both occurrence and gene abundance) when the pressure changes? Nevertheless, this question is minor, the recombinant experiment provides direct evidence for the *nosZ* activity under HHP. Fig.5A is hard to read, I would suggest separate the transcripts abundance data from the tree.

Response: We checked the presence of gene redundancy by dereplicating the sequences

at 95% identity using the mmseqs2 tool. This ended with nine *Sec-nosZ* and 22 *Tat-nosZ* clusters, therefore confirming the presence of gene redundancy. The results showed that sequence redundancy is more prevalent in *Tat-nosZ*, with up to six different *Tat-nosZ* clusters being active at the same time under 90 MPa, more than any other pressures. These clusters also belong to different microbial groups. Additionally, gene redundancy in *Sec-nosZ* was also observed with three different clusters being active at the same time under 115 MPa.

We have modified Fig. 5A and separated the transcripts abundance data from the phylogenetic tree. The detailed results (TPM) are now present in the modified Supplementary Table 5.

L173: O₂ consumption data is missing. Or is this sentence related to Extended fig1?

Response: The oxygen was continuously supplied at the same concentration to the flowing incubation system (DOES), thus was not quantified. The conclusion was based on the expression of genes associated with the aerobic respiration chain but not the dissolved oxygen concentration rate. We have replaced “O₂ consumption” with “aerobic respiration” to better describe our results in the revised manuscript L244.

L180-181: this sentence is misleading. Does this mean the denitrifiers can also use O₂ or denitrifier can live under oxic conditions while others are using O₂?

Response: By genome-centric analysis, our results showed that aerobic respiration-related genes and denitrification-related genes within the same MAG were transcribed concurrently under higher pressures. Thus, we proposed that denitrifiers (such as *Halomonas titanicae*, *Marinobacter hydrocarbonoclasticus* and *Thalassospira xiamenensis*) could simultaneously use both oxygen and nitrate as electron acceptors. This is revised as in L251-255.

L222: 30.3 nmol N /day cm³. How is this value calculated? From table S3, the N₂O production rate is 0.287 μmol/day under 70MPa, $0.287 \times 2 \times 1000 / 30.3 = 18 \text{ cm}^3(\text{ml})$, $18 \text{ ml} / 0.3 \text{ ml}/\text{min} = 60 \text{ min}$ (1H), I hope my calculation is wrong, otherwise the N₂ production

is equivalent to Atacama Trench.

Response: It is difficult to estimate the nitrogen loss rate as the N₂ production was not quantified. Therefore, we have used the calculation suggested by the reviewer and used the gaseous N₂O production rate to estimate the nitrogen loss potentials of the sediment. We rephrased the sentence as “Under laboratory conditions, we observed that the rate of nitrogen loss (N-N₂O) caused by denitrification-derived N₂O alone was approximately ~ 4.6 nmol N/day/mL under 70 MPa with 125 mL of Mariana Trench sediment as the initial inoculum (see Methods).” in the revised version L301-306. The corresponding calculation method was added in the revised manuscript L358-365 and the modified Supplementary Table 3.

L 247-249: contextual data for sampling site is vague. What is the NO₃ concentration of ambient water, how deep was the sediment sample collected, top 5cm?

Response: This information is added in the revised manuscript L330-333. The above 20-cm layers of sediments were used as the inoculum for the incubation. The concentration of NO_x⁻ (NO₂⁻ and NO₃⁻) in the overlying water sample was 213.7 µg/L (1.98 µmol/L) (referenced from Jing et al. 2018, doi:10.3389/fmicb.2018.02821).

Overall, I would suggest the authors focus more on the physiological response of denitrifiers to HHP rather than getting back to ‘in-situ’ biogeochemical mechanisms as the pre- and after- incubations drives too far away from real scenarios.

Response: We agree. The revised manuscript focuses on the physiological response of denitrifiers to HHP, instead of an *in situ* simulation as proposed in the previous version.

Reviewer #2 (Remarks to the Author):

The manuscript entitled “High hydrostatic pressure stimulates microbial nitrate reduction in hadal trench sediments under oxic condition” by Yang et al. investigated the microbiology and omics of a marine sediment sample collected at a depth of over 6,000 m. This work demonstrated that denitrification is a major metabolic process in the nitrogen cycle under aerobic conditions and high hydrostatic pressure. The results are interesting, well presented and documented. However, this work lacks control experiments (e.g. in the absence of oxygen) in my view. The experimental design is inadequate, as samples pressurized and maintained at in situ hydrostatic pressure during collection and storage, were directly incubated first at atmospheric pressure and then successively at increasing pressures applied during incubation. The enrichment of microorganisms at atmospheric pressure undoubtedly led to growth and an increase in the ratio of non-piezophilic or piezotolerant microorganisms, which became dominant during incubation compared with microorganisms adapted to in situ pressure conditions.

Response: We are glad to see that the reviewer appreciates the presentation of our results. The manuscript has been modified according to the reviewers' comments.

This study aims to investigate the effect of increased hydrostatic pressures on the microbial nitrate reduction pathway under oxygenic conditions. We don't think the anaerobic control is required since the surface sediment in the hadal trenches is typically aerobic. Having the incubation performed at anaerobic conditions may further enhance the denitrification process, but this is beyond the scope of the study. We agree that the microbial community structure changed with the sequential incubation experiment. However, due to the relatively short incubation time (15 days) at each pressure, the relative abundance of non-piezotolerant microorganisms enriched at low pressure was generally low. More importantly, the dominant denitrifier *Halomonadaceae* was relatively stable with a high abundance (23.6–37.4%) throughout the flowing incubation period (Fig. 2C and Supplementary Table 2). Thus, we believe that our experiment was sufficient to support our two main findings, which were 1) aerobic respiration and anaerobic metabolic pathways (denitrification) could occur simultaneously, and 2) microbial nitrogen metabolism route changed with increasing hydrostatic pressure.

Metagenomic and metatranscriptomic analysis lacks analysis of gene diversity and binning in relation to the dominant taxonomic groups present. These results do not support the authors' conclusions regarding active and major metabolisms *in situ*.

Response: We have performed additional genome-centric approaches and added the in-depth analysis at the MAGs level. This added content describes the dominant taxonomic groups that were actively involved in the denitrification pathway. This includes a new descriptive paragraph (L167-196), new figures (Fig. 4B and Extended Data Fig. 4), and Supplementary Tables 7 and 8 in the revised manuscript.

In addition, we also added the metatranscriptomic analysis on *in situ* RNA-fixed surface sediments from the Mariana Trench, which further supported the conclusion of simultaneous denitrification and aerobic respiration under high hydrostatic pressure in natural environments. This content is now shown as a new descriptive paragraph in the revised manuscript L217-242.

Specific comments:

- Lines 86-88: Under your experimental conditions, can you tell us what role Thaumarchaea play in the nitrogen cycle?

Response: The 16S rRNA genes of Thaumarchaea were detected in amplicon sequencing results. However, the marker gene of ammonium oxidation (*amoABC* gene) was not identified in the metagenome nor the metatranscriptome. We propose that this may be due to insufficient sequencing depth. As the cell number of archaea is two orders of magnitude lower than bacteria, the *amoABC* gene may fail to be detected (Fig. 2B). Therefore, we acknowledge the presence of Thaumarchaea, but their ammonia-oxidizing activity may be neglectable in our setup.

- Lines 112-125: Could you indicate to which taxonomic group the genes described in this paragraph can be assigned?

Response: We have carried out additional genome-centric analysis. The results showed that denitrification-associated genes were identified from 40 MAGs, and actively

transcribed in 31 MAGs under different pressures. These MAGs mainly belonged to Proteobacteria (28 MAGs), Bacteroidota (8 MAGs), Actinobacteriota (3 MAGs) and Thermoproteota (1 MAG). We found that the relative abundance of denitrifiers changed under various hydrostatic pressures (Fig. 4B). Specifically, *Marinobacter hydrocarbonoclasticus* was dominant at 40 MPa with its abundance decreased under increased pressures, but the transcriptional level of its denitrification-related genes (*nirS*, *norBC*, and *nosZ*) increased under the elevated pressures. The abundance of *Thalassospira xiamenensis* increased as pressures increased, and its genes associated with denitrification (*napAB*, *nirS*, *norBC*, and *nosZ*) were only detected as being active under 70, 90 and 115 MPa. *Idiomarina loihiensis* also participated in denitrification, in which the *nirK* and *norB* genes were actively transcribed during incubation, however, its abundance decreased with the elevated pressures. The relative abundance of *Aequorivita vladivostokensis* remained stable during incubation, but its denitrification-related genes (*nirK*, *norB*, and *nosZ*) were transcribed only under 115 MPa (Supplementary Tables 7 and 8). Lastly, *Halomonas titanicae* was the most abundant denitrifier across the entire incubation period, and its denitrification-related genes (*narGHI*, *norBC* and *nosZ*) were more actively transcribed under higher hydrostatic pressures, especially under 70, 90 and 115 MPa.

This content to describe the key taxonomic groups that are actively involved in the denitrification pathway under different hydrostatic pressures, has been as a new descriptive paragraph (L167-196), new figures (Fig. 4B and Extended Data Fig. 4), added Supplementary Tables 7 and 8 in the revised manuscript.

- Line 142: Similarly, to which taxonomic group does the selected *nosZ* gene refer?

Response: Based on the taxonomic annotation, we found that most Sec-dependent *nosZ* genes were affiliated with the Flavobacteriales, whereas the Tat-dependent *nosZ* genes were mainly affiliated with Alphaproteobacteria and Gammaproteobacteria.

This content has been added to the revised manuscript L145-147 and the modified Supplementary Table 5.

Furthermore, we also analyzed the MAGs for actively transcribing the *nosZ* genes in this

incubation system. The results showed that the Sec-dependent *nosZ* gene mainly belonged to *Aequorivita vladivostokensis* (Flavobacteriales) and *Muricauda sp002167435* (Flavobacteriales). The Tat-dependent *nosZ* gene mainly belonged to *Halomonas titanicae* (Gammaproteobacteria), *Marinobacter hydrocarbonoclasticus* (Gammaproteobacteria) and *Thalassospira xiamenensis* (Alphaproteobacteria).

This content has been shown in added Supplementary Table 8.

- Lines 180-181: this conclusion is not supported by the results obtained, i.e., what is the diversity of respiratory chains present and active, for example?

Response: This sentence is revised as “By genome-centric analysis, our results showed that aerobic respiration-related genes and denitrification-related genes within the same denitrifier MAGs, such as *Halomonas titanicae*, *Marinobacter hydrocarbonoclasticus* and *Thalassospira xiamenensis*, were transcribed concurrently under higher pressures (Fig. 6A).” as revised in L251-255.

In the revised manuscript, we additionally analyzed the MAGs that are involved in the denitrification and/or aerobic respiration pathways. The results showed that both aerobic respiration-related genes (e.g., *Cyo*, *Cyd*, *Cco*, and *Cox*) and denitrification-related genes in the dominant taxa *Halomonas titanicae* and *Marinobacter hydrocarbonoclasticus* concurrently transcribed actively under higher hydrostatic pressures (newly added Supplementary Table 9).

Moreover, we also added the metatranscriptomic data from *in situ* RNA-fixed sediments at the Mariana Trench. The results showed that the related genes (e.g., *Cyo*, *Cyd*, *Cco*, and *Cox*) encoding terminal oxidases and the key genes involved in denitrification co-occurred and both were active transcriptions (newly added Supplementary Table 10).

This content has been added to the revised manuscript L217-242.

- Lines 247—249: What are the physicochemical characteristics of the sediment sample used in this study?

Response: The amount of sediment we received in this study was insufficient to quantify the physicochemical characteristics. However, we have obtained physicochemical parameters (temperature, salinity, and nutrients) in the overlying water, which was

collected tens of centimeters above this sediment surface (Jing et al. 2018). The concentration of NO_x^- (NO_2^- and NO_3^-) was 213.7 $\mu\text{g/L}$ (1.98 $\mu\text{mol/L}$), the concentration of NH_4^+ was 101.31 $\mu\text{g/L}$ (5.63 $\mu\text{mol/L}$), the concentration of PO_4^{3-} was 82.71 $\mu\text{g/L}$ (0.87 $\mu\text{mol/L}$), the temperature was 1.59°C, the salinity was 34.69 psu (Jing et al. 2018).

Thus, we have added this information in the revised manuscript L332-333 with an appropriate citation.

In addition, the physicochemical characteristics of sediment from the Mariana Trench in previous literature are: the NO_3^- concentration was approximately 30–40 $\mu\text{mol/L}$ in the 0–20 cm layer of sediment (Nunoura et al. 2018). The O_2 concentration was greater than 100 $\mu\text{mol/L}$ in the 0–20 cm layer of 6018 m sediment (Glud et al. 2013).

- Line 254: Why was glucose used as a source of carbon and energy?

Response: We followed the protocol of Zhong & Jia, 2013; Qin et al. 2017, in which glucose was used as an organic carbon source and electron acceptor to study the nitrate reduction process of microorganisms. This has been added to the revised manuscript L338-340.

References:

Glud, R. et al. High rates of microbial carbon turnover in sediments in the deepest oceanic trench on Earth. *Nat. Geosci.* **6**, 284–288 (2013).

Jing, H. et al. Particle-Attached and Free-Living Archaeal Communities in the Benthic Boundary Layer of the Mariana Trench. *Front. Microbiol.* **9**, 2821 (2018).

Nunoura, T. et al. Microbial diversity in sediments from the bottom of the challenger deep, the Mariana Trench. *Microbes Environ.* **33**, 186–94 (2018).

Qin, Y. et al. Effect of glucose on nitrogen removal and microbial community in anammox-denitrification system. *Bioresour. Technol.* **244**, 33-39, (2017).

Zhong, Y. M. & Jia, X. S. Simultaneous ANAMMOX and denitrification (SAD) process in batch tests. *World J. Microbiol. Biotechnol.* **29**, 51-61, (2013).

REVIEWERS' COMMENTS

Reviewer #1 (Remarks to the Author):

The authors have addressed all my questions and the manuscript has been improved. The following aspects could be considered for better accuracy:

Line 23-24 'our findings demonstrate that hydrostatic pressure can restructure the chemical zonation in hadal trenches', this is from the discussion in line 275-290, no direct evidence to support 'restricting' in the hadal sediment , consider rephrasing and tone down.

Line 316-317 "High hydrostatic pressures promote denitrification activity under oxic conditions" not for all denitrifiers, but only for those survived in the depressurizing-repressurizing cycle in the chemostat. The chemostat incubation helps to recover a group of denitrifiers that can use both O₂ and nitrate but their contribution to 'in-situ' denitrification activity is undetermined.

Response to the reviewers' comments:

Reviewer #1 (Remarks to the Author):

The authors have addressed all my questions and the manuscript has been improved. The following aspects could be considered for better accuracy:

Line 23-24 'our findings demonstrate that hydrostatic pressure can restructure the chemical zonation in hadal trenches', this is from the discussion in line 275-290, no direct evidence to support 'restricting' in the hadal sediment, consider rephrasing and tone down.

Response: Thanks for your suggestion. We have rephrased and toned down this sentence, as follows: 'Taken together, our findings demonstrate that hydrostatic pressure can influence microbial contributions to nitrogen cycling and the hadal trenches are a potential nitrogen loss hotspot', in Line 28-30.

Line 316-317 "High hydrostatic pressures promote denitrification activity under oxic conditions" not for all denitrifiers, but only for those survived in the depressurizing-repressurizing cycle in the chemostat. The chemostat incubation helps to recover a group of denitrifiers that can use both O₂ and nitrate but their contribution to 'in-situ' denitrification activity is undetermined.

Response: Thanks for your suggestion. We have revised the sentence as "We observed that high hydrostatic pressure promotes denitrification activity even under oxic conditions, presenting an indication that hydrostatic pressure has the potential to modify the niche breadth and the activity of microorganisms. This modification, in turn, influences their contributions to the elemental cycling processes within hadal trenches", in Line 315-319.